# Wisconsin microbiome study, a cross-sectional investigation of dietary fibre, microbiome composition and antibiotic-resistant organisms: rationale and methods

Shoshannah Eggers,[1] Kristen MC Malecki,[1] Paul Peppard,[1] Julie Mares,[2] Daniel Shirley,[3] Sanjay K Shukla,[4] Keith Poulsen,[5] Ronald Gangnon,[1,6] Megan Duster,[3] Ashley Kates,[3] Garret Suen,[7] Ajay K Sethi,[1] Nasia Safdar[3,8]

For numbered affiliations see end of article.

**Correspondence to**
Dr Nasia Safdar;
ns2@medicine.wisc.edu

## ABSTRACT

**Introduction** Prevention of multidrug-resistant organism (MDRO) infections, such as those caused by methicillin-resistant *Staphylococcus aureus*, vancomycin-resistant enterococci, fluoroquinolone-resistant Gram-negative bacteria and *Clostridium difficile* is crucial. Evidence suggests that dietary fibre increases gut microbial diversity, which may help prevent colonisation and subsequent infection by MDROs. The aim of the Winning the War on Antibiotic Resistance (WARRIOR) project is to examine associations of dietary fibre consumption with the composition of the gut microbiota and gut colonisation by MDROs. The secondary purpose of the study is to create a biorepository of multiple body site specimens for future microbiota research.

**Methods and analysis** The WARRIOR project collects biological specimens, including nasal, oral and skin swabs and saliva and stool samples, along with extensive data on diet and MDRO risk factors, as an ancillary study of the Survey of the Health of Wisconsin (SHOW). The SHOW is a population-based health survey collecting data on several different health determinants and outcomes, as well as objective body measurements and biological specimens. WARRIOR participants include 600 randomly selected Wisconsin residents age 18 and over. Specimens are screened for MDRO colonisation and DNA is extracted for 16S ribosomal RNA-based microbiota sequencing. Data will be analysed to assess the relationship between dietary fibre, the gut microbiota composition and gut MDRO colonisation.

**Ethics and dissemination** The WARRIOR project is approved by the University of Wisconsin Institutional Review Board. The main results of this study will be published in a peer-reviewed scientific journal.

### BACKGROUND

Trillions of microorganisms colonise the human body and play an important role in our health by affecting metabolism, nutrition, immune function and nervous system signalling.[1] Given their association with

> **Strengths and limitations of this study**
>
> ► This study uses a large, non-clinical, population-based sample with a wide variety of exposures to multidrug-resistant organism risk factors.
> ► The extensive data and biological specimens collected by the Survey of the Health of Wisconsin and the Winning the War on Antibiotic Resistance project allow for future use in many more studies examining a variety of different hypotheses.
> ► The primary limitation of this study is its cross-sectional nature; however, plans for follow-up data collection are underway.

these varying biological mechanisms, imbalance or dysbiosis of the gut microbiota has been linked to many adverse health effects including increased risk for infection, obesity, diabetes, inflammatory bowel disease, allergic disease, frailty in ageing and mental health conditions.[1 2] There is no consensus on what microbial composition constitutes a healthy gut microbiota, although a more diverse microbiota is thought to be better, especially in the case of healthy immune response and protection against infection.[3]

Infection with multidrug-resistant organisms (MDROs) is increasingly common and effective treatment options are rapidly decreasing.[4] Vancomycin-resistant enterococci (VRE), fluoroquinolone-resistant Gram-negative bacilli (FQRGNB), methicillin-resistant *Staphylococcus aureus* (MRSA) and *Clostridium difficile* are all MDROs with the capacity to cause seriously detrimental health effects.[5] VRE often causes infections associated with hospitalisation, including urinary, bloodstream, catheter and surgical wound infections.[6] FQRGNB can cause pneumonia,

sepsis, meningitis and surgical site infections.[7] *S. aureus* is carried by approximately 30% of the US population, while MRSA is carried by about 1%.[8] *S. aureus* carriage can be commensal but leads to increased risk for infection by MRSA.[9] *C. difficile* causes more than 450 000 infections, leading to 15 000 mortalities annually and has exceeded MRSA as the most frequent cause of hospital-acquired infection.[10 11] The lack of effective treatment options for these infections also endangers the efficacy and outcomes of other medical treatments, including surgery and those for cancer.[12] MDROs are often transmitted in healthcare settings but are increasingly being acquired through community sources.[13] In addition to causing clinical disease, MDROs can cause asymptomatic colonisation which is a strong predictor of future infection[14] and can be a source of transmission via asymptomatic carriers of MDROs.[15] Preventing colonisation by MDROs is therefore vital to preventing infection.

A balanced microbiota can prevent colonisation and infection with MDROs and other pathogens via several pathways. One mechanism is competitive inhibition, whereby commensal microbes compete for the same resources and mucosal binding sites as pathogenic bacteria and limit their growth.[16] The makeup of the microbiota also plays a large role in the development of the immune system and continues to influence immune response and maintain homeostasis throughout our lives.[17] Beneficial bacteria within the microbiota produce cytokines, short and long-chain fatty acids and other signalling molecules that increase mucus production and strengthen epithelial barriers, as well as increasing type 1 T helper cell response, all of which help to fight off pathogenic bacteria.[18]

Many factors are known to influence the composition of the human gut microbiota, including age, sex and genetics, as well as modifiable factors including birth mode, diet, exercise, environment, smoking, cohabitation, animal contact and use of antibiotics, probiotics and prebiotics.[19–23] Recent literature suggests dietary factors can alter the gut microbiota and may play a role in the risk of infection by gut pathogens.[24] Dietary fibre appears promising in promoting a diverse, healthy gut microbiota by selecting for fibre-degrading microbes that produce immune-enhancing compounds like butyrate.[25] Butyrate and other short-chain fatty acids are end-products of microbial fermentation that can enter systemic circulation and inhibit the expression of specific proinflammatory cytokines.[26] Moreover, disease-causing disturbances to the gut microbiota may be due to Western diets abundant in fats and simple carbohydrates but lacking in fibre.[27] Although these links between fibre and immune function via the gut microbiota are promising, there is a paucity of data on the relationship of fibre with colonisation resistance against MDROs, particularly in non-clinical populations.

The purpose of the Winning the War on Antibiotic Resistance (WARRIOR) study is to examine the relationships between dietary fibre, the gut microbiota and colonisation by MDROs in a statewide, non-clinical, population-based sample of adults and to further create a microbiome sample repository for future research. We aim to determine the association between diets either high or low in fibre and gut microbial diversity to examine the different effects of specific types of dietary fibre on the gut microbiota and MDRO colonisation. The primary hypothesis is that higher dietary fibre consumption will be associated with increased gut microbial diversity and lower prevalence of MDRO colonisation.

## METHODS/DESIGN
### Overview
The WARRIOR project aims to collect data and biological samples from 600 Wisconsin residents age 18 and over. WARRIOR is an ancillary study of the ongoing Survey of the Health of Wisconsin (SHOW), for which methods have been previously published.[28] A description of the WARRIOR project and the full SHOW protocol are available on the SHOW website (www.show.wisc.edu). The SHOW is an annual cross-sectional, statewide, population-based health survey, modelled after the National Health and Nutrition Examination Survey, which collects a wide range of health, behaviour and environment data as well as objective body measurements and biological specimens. The SHOW was initiated in 2008 and the WARRIOR project is a 2-year ancillary study that began at the start of the 2016 survey year. Survey components that were added to the SHOW by the WARRIOR project include additional dietary assessments, questions about MDRO risk factors and additional specimen collection including swabs of oral, skin and nasal tissues, as well as saliva and stool samples. A study schematic outlines the various study components in figure 1.

### Recruitment and compensation
Subjects are enrolled for the WARRIOR project during the SHOW recruitment and complete the WARRIOR project components in addition to the SHOW survey components. The SHOW participants age 18 and over, all of whom meet the inclusion and exclusion criteria listed in box 1, are invited to participate in the WARRIOR project. Participants complete an informed consent for both the SHOW and WARRIOR components, as approved by the University of Wisconsin-Madison Institutional Review Board. Participants are compensated for each component of the survey that they complete.

### Dietary assessment
The WARRIOR project added two dietary assessments, in addition to those already included in the SHOW, that allow for the assessment of usual total fibre intake and fibre from different sources, and intake of macronutrients, phytochemicals, vitamins and minerals. Usual diet over the past year is queried using the National Cancer Institute's Diet History Questionnaire II.[29] The second added dietary component is an Automated Self-Administered

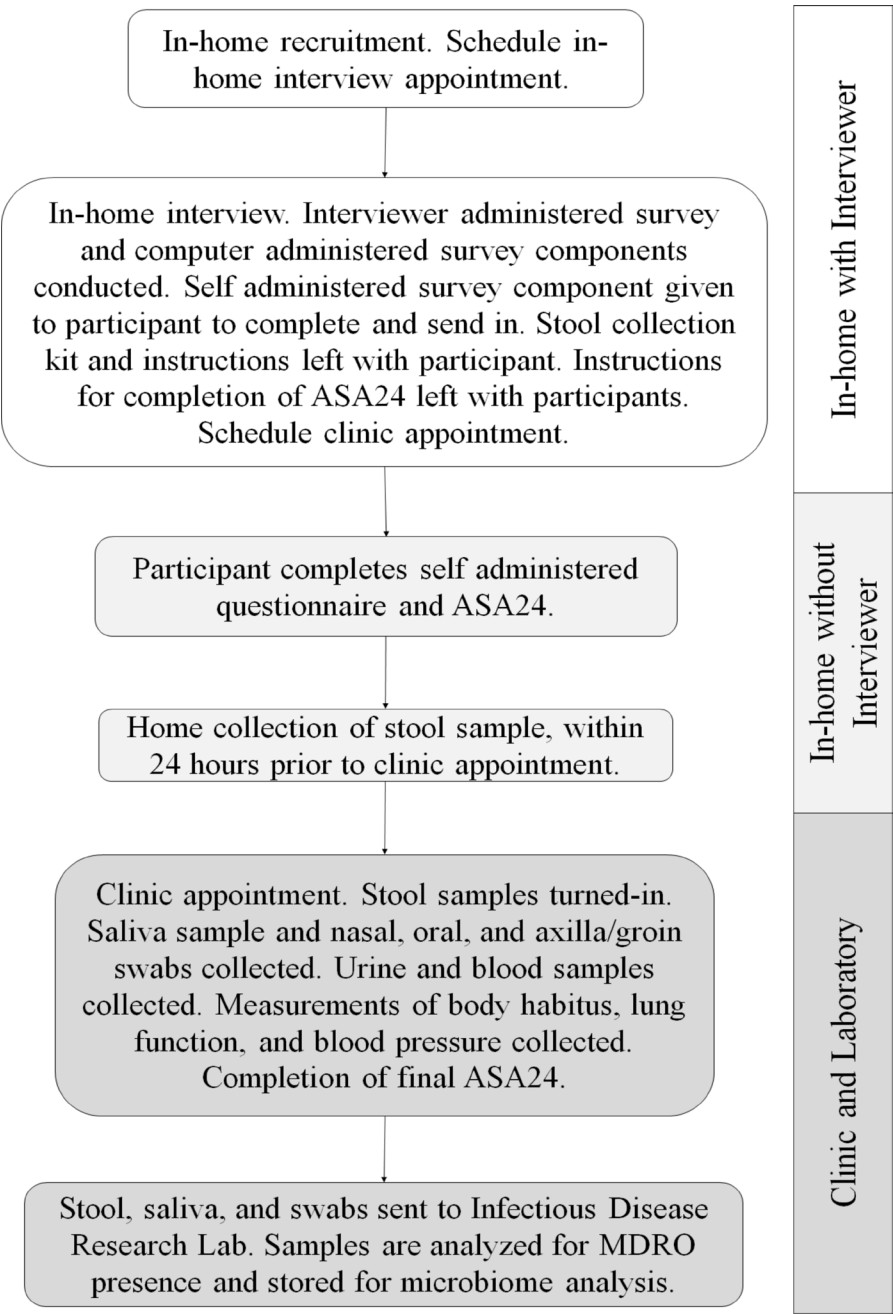

**Figure 1** A study schematic outlining components of data and specimen collection. ASA24, Automated Self-Administered 24 hours Dietary Assessment; MDRO, multidrug-resistant organism.

24 hour Dietary Assessment (ASA24)[30] completed online by participants, which queries intake over a 24-hour period. Dietary fibre intake will be assessed for statistical analysis by average daily grams of consumption.

When the WARRIOR project started, participants were asked to complete the ASA24 four times. Completion of the ASA24 was found to be difficult for many participants due to lack of a reliable internet connection, as well as the length and complexity of the assessment. Completion of all four ASA24s added significantly to participant survey fatigue, and completion rates were 21% for 1 recall, 23% for 2 recalls and 16% for 3 or 4 recalls after the first 5 months. Ultimately, our protocol

was modified to request the completion of the ASA24 twice, at appointments where there are computers and personnel assistance for online completion. Participants are compensated for attempting to complete at least one ASA24.

### MDRO risk factor assessment

Several risk factors for MDRO colonisation, outlined in the conceptual model illustrated in figure 2, were incorporated into the SHOW's interview and questionnaire components (see online supplement 1). Given the novelty of this study, standard questionnaires assessing exposure to MDRO risk factors were not readily available.

**Box 1**   **List of inclusion and exclusion criteria for participation in the WARRIOR project**

**Inclusion criteria**
► The selected household is the individual's usual place of residence.
► Age 18 years or older.
► Mentally capable of giving written informed consent.
► Able to communicate answers to interview question.

**Exclusion criteria**
► Residents of nursing homes, hospitals, mental institutions, penal institutions, jails, halfway houses or who are under the jurisdiction of the Department of Corrections.
► Students not currently residing in the selected residence.
► Full-time members of the armed forces or activated units of the National Guard who are currently stationed away from home and do not usually sleep in the residence.
► Individuals who are visiting the household.
► Individuals who have two residences and spend the greater number of nights at the other residence.
► Individuals who have voluntarily disclosed a diagnosis of mental incapacity.

Thus, questions were developed by the WARRIOR project team, a group with wide-ranging expertise in microbiology, epidemiology, infectious disease and nutrition. Questions were piloted to evaluate face validity. Exposure to domestic and farm animals are assessed because they can carry MDROs and can affect non-pathogenic components of the microbiome. We ask about farm exposure, where MDROs are often present, particularly among livestock and the use of antibiotics and proton pump inhibitors, which can have substantial and direct effects on the bacteria within the microbiome by selecting for antibiotic resistance. Questions about exposure to hospitals and history of MDRO infection, both important predictors of future MDRO infection, are also included. All SHOW and WARRIOR questionnaires and data codebooks are available at https://www.med.wisc.edu/show/data-service-center/, and MDRO risk factor assessment instruments can been found in online supplement 1. Because these questions are distributed throughout the existing SHOW components, they did not suffer noticeably different completion rates from SHOW components.

## Biological sampling

In addition to the blood and urine specimens collected by the SHOW, the WARRIOR project collects saliva and stool samples and separate swabs of the nose, mouth and skin (combined axilla/groin). Participants self-collect a stool sample at home using a collection kit provided by the SHOW interviewer that includes a stool collection hat, a sterile 60 mL specimen cup, a sterile wood tongue depressor, gloves, a specimen label, a biohazard bag, a brown paper bag and an instruction sheet. Participants collect the stool sample within the 24 hours prior to their SHOW clinic visit and refrigerate the sample until submitting it at their appointment. At the clinic appointment approximately 1–2 mL of saliva is collected using a sterile collection aid and a sterile tube, and swabs of the axilla/groin, nares and buccal mucosa and tonsils are taken using a dual head BBL CultureSwab with liquid stuart transport medium (Becton, Dickinson and Company, Franklin Lakes, New Jersey, USA). All WARRIOR samples are then shipped and received at the Infectious Disease Research Laboratory at the University of Wisconsin—Madison within 24 hours, where they are immediately processed for MDRO colonisation testing and then frozen at −80°C for later use in microbiome analysis. While stool collection and shipment proved to be easier for participants than anticipated, saliva collection was more inconsistent than expected, as ease and rate of saliva production can vary greatly among individuals.

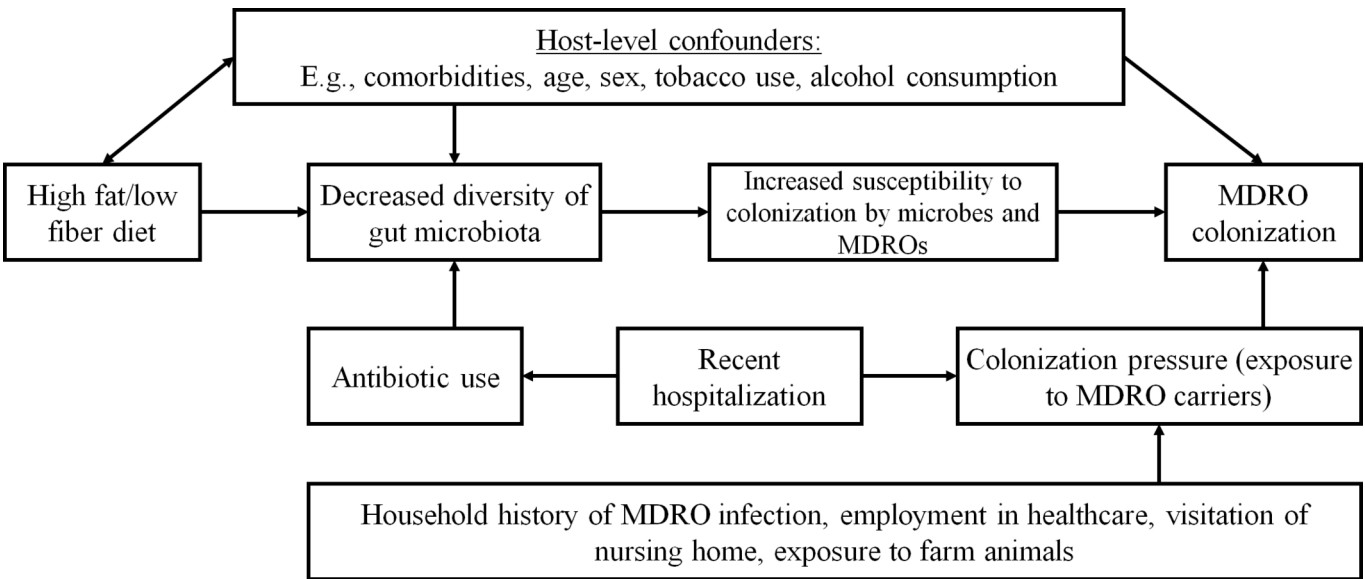

**Figure 2**   A conceptual model illustrating the pathways between dietary fibre consumption and MDRO colonisation, including mediators and confounding factors. MDRO, multidrug-resistant organism.

## Microbiological analysis

In 2016, swabs, saliva and stool were screened for the presence MRSA, VRE and FQRGNB; in 2017 screening for *C. difficile* was added. Specimens are processed immediately on receipt by the laboratory. Swabs are vortexed in 1 mL of tryptic soy broth (TSB) (Remel, Lenexa, Kansas, USA) while 100 µL of saliva and 0.1 g of stool are used to inoculate the TSB, resulting in a total of five assays per subject that completes all biological components of the WARRIOR project. Broths are incubated overnight aerobically at 36°C. Aliquots of each broth are plated to mannitol salt agar (Remel, Lenexa, Kansas, USA) supplemented with 4 mg/L of cefoxitin (Sigma-Aldrich, St Louis, Missouri) for MRSA detection,[31] enterococcosel agar (BD/Difco, Sparks, Maryland) supplemented with 6 mg/L of vancomycin (Sigma-Aldrich, St Louis, Missouri) for VRE detection and MacConkey's agar (BD/Difco, Sparks, Maryland, USA) supplemented with 4 mg/L of ciprofloxacin (Sigma-Aldrich, St Louis, Missouri) for detection of FQRGNB. Colonies matching suspected morphology on selective agar are subcultured on blood agar plates (BAP) (BD, Sparks, Maryland) for identification. Identification of isolates is performed using conventional biochemical methods and identification is confirmed via sequencing of the 16S ribosomal RNA (rRNA) gene. Resistance to cefoxitin and ciprofloxacin are determined using Kirby-Bauer disc diffusion susceptibility testing methods and breakpoints published in the Clinical Laboratory Standards Institute documents M07-A10 and M100-S26.[32 33] The E-test (Bio-Merieux, Marcy l'Etoile, France) is used to determine the minimum inhibitory concentration (MIC) of vancomycin. For the added *C. difficile* detection, 0.1 g of stool is inoculated into 1 mL of prereduced *C. difficile* Brucella Broth and incubated anaerobically at 36°C overnight.[34] Fifty microlitres is plated on a *C. difficile* Brucella agar plate and incubated for 48 hours at 36°C. Colonies matching suspected colony morphology are subcultured to a prereduced BAP and subsequently identified using Gram staining and catalase testing. Presence of toxin genes is assessed using an in-house PCR assay and bacterial identification is confirmed via sequencing of the 16S rRNA gene.[35] All positive antibiotic resistant isolates are stocked and stored at −80°C for future unspecified research.

Microbiota analysis is performed using DNA extracted and purified from stool samples to address the aims of the WARRIOR project, and DNA extracted from other sample matrices will be used for future unspecified research. The purified DNA is then normalised to a concentration of 5 ng/µL and amplified using PCR with barcoded primers to the 16S V4 region and sequenced on an Illumina Miseq (2×250 bp reads).[36] Stored DNA samples are available as a resource for additional metagenomic research and additional analyses as new technologies are developed.

## Stool genomic DNA extraction

Approximately 180–220 mg of each faecal sample is added to a 2 mL bead-beating tube containing 500 µL 2× sodium chloride-tris-EDTA (STE) buffer, 300 mg of 1.0 mm diameter zirconia/silica beads and vortexed to homogenise the stool. The sample is then centrifuged for 15 min at 4°C at 500g. A total of 800 µL of 2× STE buffer is added to the supernatant and up to 1000 µL is transferred to a new bead-beat tube containing 0.1 mm diameter zirconia/silica beads and one 4 mm stainless steel bead. For chemical lysis, 115 µL of an enzymatic cocktail containing 50 µL lysozyme (10 mg/mL), 10 µL mutanolysin (1 mg/mL), 5 µL lysostaphin (5 mg/mL) and 50 µL 20% sodium dodecyl sulfate is added to each tube. Additionally, 700 µL phenol:chloroform:isoamyl alcohol is added to the sample. Bead-beat tubes are then vortexed and incubated at 56°C for 30 min. For mechanical lysis, bead-beat tubes are vortexed and then placed in a Mini-BeadBeater-24 (Cat 112011, Biospec Products, Bartlesville, Oklahoma, USA) and beat for 3 min. Tubes are centrifuged at 16 000xg for 10 min at 4°C. The top aqueous layer is transferred to a clean 2 mL tube and washed with an additional 500 µL phenol:chloroform:isoamyl alcohol and vortexed. The sample is then centrifuged at 16 000×g for 10 min at 4°C. The phenol:chloroform:isoamyl alcohol wash is repeated between 2 and 10 times to remove impurities from the sample until the aqueous layer is clean. The top aqueous layer is then transferred to a clean 2 mL microcentrifuge tube containing 70 µL of 3M sodium acetate and 700 µL isopropanol. The samples are inverted several times and subsequently incubated at −20°C for 30 min to 1 hour. Each sample is centrifuged at 16 000xg (4°C) for 20 min to collect the DNA pellet, which is then washed with 500 µL cold 70% ethanol. The ethanol wash is repeated, and sample DNA pellets are dried for 5 min using a Savant SpeedVac (DNA120-230, Thermo Scientific, Waltham, Massachusetts, USA). Finally, dried DNA pellets are resuspended in 100 µL TE buffer and stored overnight at 4°C or at 37°C for 1 hour to dissolve the DNA pellet. Samples are then purified using NucleoSpin Gel and PCR clean-up kit according to manufacturer's directions (Macherey-Nagel, Germany) and eluted in 40 µL TE buffer. DNA is quantified using PicoGreen in a microplate reader (BioTek Instruments) and stored long term at −80°C.

## Swab and saliva genomic DNA extraction

The swab head is placed into a 2 mL bead-beating tube containing 750 µL 1× PBS and 500 mg of 0.1 mm diameter zirconia/silica beads. For chemical lysis, 25 µL of an enzymatic cocktail containing 5 µL lysozyme (10 mg/mL), 15 µL mutanolysin (1 mg/mL) and 5 µL lysostaphin (5 mg/mL) is added to each bead-beat tube and vortexed. The bead-beat tubes are then incubated at 37°C for 30 min before 60 µL of a second enzymatic cocktail containing 10 µL proteinase K (20 mg/mL) and 50 µL 10% sodium dodecyl sulfate is added to each tube. Bead-beat tubes are then vortexed and incubated at 55°C for 45 min. For mechanical lysis, bead-beat tubes are vortexed and then placed in a Mini-BeadBeater-24 (Cat 112011, Biospec Products, Bartlesville, Oklahoma, USA)

and beat for 3 min. Tubes are centrifuged at 16 000 $g$ for 3 min at 4°C. The top aqueous layer is transferred to a clean 2 mL microcentrifuge tube containing 70 µL of 3M sodium acetate and 700 µL isopropanol. The samples are inverted several times and subsequently incubated at −20°C for 30 min to 1 hour. The following ethanol wash, pellet drying and resuspension, column purification, DNA quantification and storage steps are identical to those used in the stool genomic DNA extraction method above.

## Statistical considerations

The proposed sample size of 600 subjects will provide 80% power to detect a partial correlation (after adjustment for covariates) of 0.125 between dietary fibre intake and the primary outcome, a diversity index using a two-sided 2.5% level test.

Raw sequencing data will be processed using mothur.[36] Contigs (overlapping sequences) will be compiled, and low-quality reads will be removed. Sequences of short length and chimaeras will be detected and removed using UCHIME.[37] Sequences will be assigned to operational taxonomic units (OTUs) at the species level (97% similarity) using the GreenGenes database.[38] OTU counts and the diversity (Shannon and Simpson) and richness (ACE and Chao) indices will be calculated.[39–41]

Several different regression methods will be used to assess the association of the usual intake of total dietary fibre and fibre from specific sources to gut microbial diversity, as well as the relationship between fibre consumption and MDRO colonisation. For example, to assess the association between dietary fibre consumption and gut microbial diversity, least squares linear regression will estimate mean species diversity as a function of dietary fibre. Usual grams of daily dietary fibre intake will be assessed by quantiles of consumption as fits the distribution of the data. Control variables will be added sequentially in groups; initial models will adjust only for demographic factors, subsequent models will add medications, and final models will add comorbidity and other risk factor data. Each variable in the model building process will be assessed individually and variables that are not significant at the ≤0.2 level will not be included in the final model. Logistic regression models will estimate proportion of subjects colonised, dichotomised as colonised by any MDRO versus not colonised by any MDRO as a function of dietary fibre using a similar modelling strategy.

## DISCUSSION

Emergence of antibiotic resistance and MDROs are a global public health crisis. These infections are often very serious, leading to increased medical care usage and death. Gaining a better understanding of how the gut microbiota influence colonisation of MDROs will help in developing new therapeutic and prevention strategies.

This is the first statewide microbiota study assessing the relationship of MDROs and diet in a random, non-clinical, general population sample. Studies of community acquired MDROs are becoming more common; however, many of these sample from community-living facilities, daycares or within livestock workers.[42–44] This study is innovative in that it samples by household within census block groups, and participants have a wider variety of exposure levels to different community acquired MDRO risk factors.

Other than low rates of ASA24 completion, participation in the added WARRIOR project components exceeded expectations. We anticipated approximately 50% of the SHOW participants would be willing to enrol in the WARRIOR project. In the first year of recruitment however, participation rates were much higher. Most people were willing to participate by submitting one or more biological samples. Having a large part of the compensation structured around the WARRIOR project components also helped with recruitment. Incorporating the MDRO risk factor questions within the usual SHOW survey likely also helped bolster completion rates.

While this study will help us better understand the relationship of dietary fibre, the gut microbiome and MDRO colonisation and serves as a biorepository for future analysis using the other biological samples collected, there are some limitations. Dietary intake data and many confounding variables to be considered are collected by self-report, although there are important exceptions (eg, physical activity and sleep are assessed by multiday accelerometry). The current WARRIOR project protocols are cross-sectional; however, the recently funded Population-based Microbiome Research Core (PMRC)[45] will conduct longitudinal follow-up of the WARRIOR sample. PMRC will collect an additional stool sample, environmental samples and reassess MDRO risk factor exposures, including questions about infection history after the WARRIOR project. This data will be useful for many future studies, including analysis assessing infection risk in addition to MDRO colonisation analysed by the WARRIOR project.

The data collected for the WARRIOR project, in addition to the extensive SHOW data, creates a rich resource that can be used for many future studies. Future directions include investigating other components of the diet, and other exposures that may be associated with the gut microbiota and MDRO colonisation. Given the many varied biological samples taken, a variety of relationships with the oral, skin and nasal microbiota could also be examined. Further assessment of the stool samples, including metagenomics and strain typing, is also a likely future direction. The established study infrastructure provided by the SHOW also allows for the possibility of collecting additional specimens in the future, for example, environmental samples such as water and dust or additional analysis of individual-level data generated from the SHOW biorepository. The ongoing infrastructure also supports additional data collection and longitudinal follow-up using these same protocols. The WARRIOR project serves as a model for population-based microbiome research

and findings will provide important insights into human variability and the role of the microbiome in protection or exacerbation of the global MDRO crisis.

**Author affiliations**
[1]Department of Population Health Sciences, School of Medicine and Public Health, University of Wisconsin-Madison, Madison, Wisconsin, USA
[2]Department of Ophthalmology and Visual Sciences, School of Medicine and Public Health, University of Wisconsin-Madison, Madison, Wisconsin, USA
[3]Division of Infectious Disease, Department of Medicine, School of Medicine and Public Health, University of Wisconsin-Madison, Madison, Wisconsin, USA
[4]Center for Human Genetics, Marshfield Clinic Research Institute, Marshfield, Wisconsin, USA
[5]Department of Medical Sciences, School of Veterinary Medicine, University of Wisconsin-Madison, Madison, Wisconsin, USA
[6]Department of Biostatistics and Medical Informatics, School of Medicine and Public Health, University of Wisconsin-Madison, Madison, Wisconsin, USA
[7]Department of Bacteriology, College of Agricultural and Life Sciences, University of Wisconsin-Madison, Madison, Wisconsin, USA
[8]William S. Middleton Veterans Affairs Medical Center, Madison, Wisconsin, USA

**Contributors** SE made contributions to the acquisition of data and was a major contributor in writing the manuscript. KM and PP made contributions to design and acquisition of data and critically revised the manuscript for important intellectual content. JM, DS, SKS, KP, RG and GS made contributions to conception and design of the study and critically revised the manuscript for important intellectual content. MD and AK made contributions to the design and acquisition of data and were involved in drafting and revising the manuscript. AKS and NS made substantial contributions to the conception and design of the study and acquisition of the data and critically revised the manuscript for important intellectual content. All authors read and approved the final manuscript.

**Funding** This work was supported by the University of Wisconsin School of Medicine and Public Health through the Wisconsin Partnership Program

**Disclaimer** The funding source had no role in the collection, analysis or interpretation of data or the writing of this manuscript.

**Competing interests** None declared.

**Patient consent** Detail has been removed from this case description/these case descriptions to ensure anonymity. The editors and reviewers have seen the detailed information available and are satisfied that the information backs up the case the authors are making.

**Ethics approval** The SHOW and WARRIOR projects were reviewed and approved by the University of Wisconsin Institutional Review Board (2013-0251). All subjects consented to study participation.

**Provenance and peer review** Not commissioned; externally peer reviewed.

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
