## [Reviewer comments · BMJ Open]

ARTICLE DETAILS

TITLE (PROVISIONAL)	The Wisconsin Microbiome Study, a Cross-Sectional Investigation of Dietary Fiber, Microbiome Composition, and Antibiotic-Resistant Organisms: Rationale and Methods
AUTHORS	Eggers, Shoshannah; Malecki, Kristen; Peppard, Paul; Mares, Julie; Shirley, Daniel; Shukla, Sanjay; Poulsen, Keith; Gangon, RE; Duster, Megan; Kates, Ashley; Suen, Garret; Sethi, Ajay; Safdar, Nasia

VERSION 1 – REVIEW

REVIEWER	Frédéric Raymond Université Laval, Canada
REVIEW RETURNED	20-Sep-2017

GENERAL COMMENTS	This manuscript describes the WARRIOR study, which aims to document the relationship between dietary fibers, microbial diversity and colonization with multiresistant microorganisms. Overall, the study is well described and study design should allow better insight on their hypothesis, which suggests that intake of dietary fibers could led to reduced colonization by MDRO. If found unclear what would be the expected scope of the study. The authors propose to include 600 subjects, but they also suggest that multiple samples could eventually be obtained from some subjects. This is not clear and should be better explained. A schematic representation of the proposed study would be appreciated and would help understanding the experimental design, type of samples, and variables considered in the study. Inclusion and exclusion criteria should be clearly stated. The following two comments are limitations to the study that the authors should consider at should at least discuss in their manuscript: Have the authors considered sequencing the genomes of the MDRO isolates (and control sensitive isolates) ? This could provide critical information on the dissemination of MDRO isolates in their region. In addition, association between diversity of the microbiome and the risk of colonization could be strain-specific. I find it a bit unfortunate that the authors choose to rely only on 16S to evaluate microbiome composition. If they sequenced complete
--

	metagenomes, their study design would also allow to test the hypothesis of whether fiber intake affects the amount of antibiotic resistance genes found in microbial populations. Also, while 16S allows to calculate diversity indexes and determine the taxonomical content of microbiomes, fiber intake could also affect strain diversity, which will not be visible with 16S.
--	--

REVIEWER	Jessica Galloway-Pena MD Anderson Cancer Center USA
REVIEW RETURNED	28-Nov-2017

GENERAL COMMENTS	As a methods paper, my primary concern is that there is insufficient information to determine if the study design is appropriate to answer the research question presented. Specifically, the authors are critically lacking in detailing the methods specifically so that the study could be repeated by another institution, and describing the statistical methods used. Moreover, how the covariates and outcomes will be assessed is not clear. Specific comments:  1. Even discussed by the authors, colonization by MRDO does not necessarily mean infection or transmission, thus I think it is within the authors best interest to include MRDO infection follow-up of the individuals in relation to diet/microbiome factors as an outcome. 2. Regarding MRDO risk factors/questions. Since it appears the questionnaire for risk factors for MRDO is unique to this study and not universal or widely recognized (like the dietary questionnaires are) I would  a. want to see the all questions ...perhaps in a supplement b. know how the authors plan on scoring the risk for MRDO based on this questionnaire and c. know how do the authors plan to validate this questionnaire 3. The authors give very little information about how the samples are collected or what their take home collection kits include ...other than "swabs". It would be nice to know the brand/vendor of the swabs...i.e. are they self-contained like BD culture swabs? Or do they use some other kind of swab and store in their own container? Do these have stabilization buffer...like in an omnigene kit? Even shipment in under 24hrs can reduce DNA yield ... 4. Although "dietary fiber consumption" or "usual fiber intake" is one of their primary variables to be assessed, the authors give no indication on how fiber consumption will be scored or assessed other than naming the questionnaires used to collect the information. Is this assessed in frequency? Amount? What kind of variables will these be? Binary? Continuous? Categorical? 5. On the same note, they do not say how they will assess MRDO colonization, will this be a simple yes/no MRDO (binary)? Will they split the outcomes by organisms? Which may diminish their power.... 6. They also say "a diversity index" but do not specify which oneShannon? Chao? Simpson? 7. Based on what type of data covariates and outcomes, the authors should lay out what types of regression methods they will use for each type of data, as there are so many ways in which to handle this data (at least give an idea). 8. Because there are so many possible different ways to handle the covariates and what types of data they could be categorized as, it hard to discern if their power analyses is justified,
--

	or at least what it is based on. 9. Also, the authors do not detail how they have collected the information, will assess, and adjust for the host-level confounders detailed in Figure 1. If these comments can be addressed and this information detailed, this manuscript will then be suitable for publication.
--	---

VERSION 1 – AUTHOR RESPONSE

December 20, 2017

Re: bmjopen-2017-019450 / The Wisconsin Microbiome Study, a Cross-Sectional Investigation of Dietary Fiber, Microbiome Composition, and Antibiotic-Resistant Organisms: Rationale and Methods.

Dear Editors:

We would like to thank you for your consideration of this resubmission for the manuscript entitled, “The Wisconsin Microbiome Study, a Cross-Sectional Investigation of Dietary Fiber, Microbiome Composition, and Antibiotic-Resistant Organisms: Rationale and Methods,” submission number bmjopen-2017-019450. We also thank the reviewers for their thoughtful review of the original manuscript, and for the insightful comments and critiques which have been addressed in the resubmitted manuscript. Based on these suggested revisions, we feel that the manuscript is much improved. Detailed below is a point-by-point reply to reviewers’ comments. Changes in the manuscript text are shown in Track Changes.

On behalf of my co-authors, we look forward to your further review and thank you for your consideration of this manuscript.

Sincerely,

Nasia Safdar, MD, PhD
 Associate Professor
 Division of Infectious Disease
 Department of Medicine
 School of Medicine and Public Health
 University of Wisconsin - Madison

Reviewer: 1

If found unclear what would be the expected scope of the study. The authors propose to include 600 subjects, but they also suggest that multiple samples could eventually be obtained from some subjects. This is not clear and should be better explained.

- The scope of this study is 600 people. A recently funded study will go back to WARRIOR participants and collect more samples, but that is not part of the scope of this study. Details have been added to the discussion section to clarify this.

A schematic representation of the proposed study would be appreciated and would help understanding the experimental design, type of samples, and variables considered in the study.

- Schematic has been added.

Inclusion and exclusion criteria should be clearly stated.

- A table listing criteria has been added.

Have the authors considered sequencing the genomes of the MDRO isolates (and control sensitive isolates)? This could provide critical information on the dissemination of MDRO isolates in their region. In addition, association between diversity of the microbiome and the risk of colonization could be strain-specific.

- This is a very interesting point, although it is outside the scope of this study. The current budget does not allow for entire genome sequencing of positive isolates, however isolates are stored for future analysis. A sentence was added to the microbiological methods section to highlight that point. This analysis could be undertaken as its own study in the future.

I find it a bit unfortunate that the authors choose to rely only on 16S to evaluate microbiome composition. If they sequenced complete metagenomes, their study design would also allow to test the hypothesis of whether fiber intake affects the amount of antibiotic resistance genes found in microbial populations. Also, while 16S allows to calculate diversity indexes and determine the taxonomical content of microbiomes, fiber intake could also affect strain diversity, which will not be visible with 16S.

- This point is well made, however, we do not believe that metagenomics analysis is necessary to address the aims of this study. Future studies can use specimens and DNA extracted from WARRIOR samples to address the further hypothesis suggested by this reviewer, as well as many other hypotheses. A sentence has been added to the discussion section to address this point.

Reviewer: 2

As a methods paper, my primary concern is that there is insufficient information to determine if the study design is appropriate to answer the research question presented. Specifically, the authors are critically lacking in detailing the methods specifically so that the study could be repeated by another institution, and describing the statistical methods used. Moreover, how the covariates and outcomes will be assessed is not clear.

- Details (as outlined below) have been added to the methods section to improve transparency and reproducibility of the study. Statistical methods used for analysis will also be detailed in the final results manuscript once the study is complete.

1. Even discussed by the authors, colonization by MRDO does not necessarily mean infection or transmission, thus I think it is within the authors best interest to include MRDO infection follow-up of the individuals in relation to diet/microbiome factors as an outcome.

- A recently funded study (PMRC) will go back to WARRIOR participants and reassess infection history, which will allow us to investigate infection risk as part of that future study. This point has been addressed in the discussion section.

2. Regarding MRDO risk factors/questions. Since it appears the questionnaire for risk factors for MRDO is unique to this study and not universal or widely recognized (like the dietary questionnaires are) I would --

a. want to see the all questions ...perhaps in a supplement

- All SHOW/WARRIOR questionnaires are available at <https://www.med.wisc.edu/show/data-service-center/>, and a supplement has been added that includes all WARRIOR components. A sentence with link has been added to the methods section.

b. know how the authors plan on scoring the risk for MRDO based on this questionnaire and

- Each risk factor variable will be included individually in the model building process. More detail has been added to the statistics section of the methods to address this.

c. know how do the authors plan to validate this questionnaire

- Details of questionnaire development and validation have been added to the methods section.

3. The authors give very little information about how the samples are collected or what their take home collection kits include ...other than “swabs”. It would be nice to know the brand/vendor of the swabs...i.e. are they self-contained like BD culture swabs? Or do they use some other kind of swab and store in their own container? Do these have stabilization buffer...like in an omnigene kit? Even shipment in under 24hrs can reduce DNA yield ...

- Details of the specimen collection tools have been added to the methods section.

4. Although “dietary fiber consumption” or “usual fiber intake” is one of their primary variables to be assessed, the authors give no indication on how fiber consumption will be scored or assessed other than naming the questionnaires used to collect the information. Is this assessed in frequency? Amount? What kind of variables will these be? Binary? Continuous? Categorical?

- Details have been added to the Dietary Assessment and Statistics sections of the methods to address this comment.

5. On the same note, they do not say how they will assess MRDO colonization, will this be a simple yes/no MRDO (binary)? Will they split the outcomes by organisms? Which may diminish their power....

- The MDRO outcome will be binary. Details have been added to the statistics section of the methods.

6. They also say “a diversity index” but do not specify which oneShannon? Chao? Simpson?

- Shannon, Simpson, ACE and Chao will all be calculated. Details and references have been added to the statistics section of the methods.

7. Based on what type of data covariates and outcomes, the authors should lay out what types of regression methods they will use for each type of data, as there are so many ways in which to handle this data (at least give an idea).

- Details of the regression analysis planned have been added to methods section.

8. Because there are so many possible different ways to handle the covariates and what types of data they could be categorized as, it is hard to discern if their power analyses is justified, or at least what it is based on.

- Details of the regression analysis planned have been added to methods section. Power calculations were based on the specified analysis.

9. Also, the authors do not detail how they have collected the information, will assess, and adjust for the host-level confounders detailed in Figure 1.

- The supplemental questionnaires as well as statistical analysis details added should address this concern.